# Predictive Factors for Skip Lymph Node Metastasis and Their Implication on Recurrence in Papillary Thyroid Carcinoma

**DOI:** 10.3390/biomedicines10010179

**Published:** 2022-01-16

**Authors:** Young-Jae Ryu, Seong-Young Kwon, Soo-Young Lim, Yong-Min Na, Min-Ho Park

**Affiliations:** 1Department of Surgery, Chonnam National University Medical School, 322 Seoyang-ro Hwasun-eup, Hwasun-gun, Gwangju 58128, Korea; brandon-surgery@hotmail.com (Y.-J.R.); mmangomani44@gmail.com (S.-Y.L.); nadoc84@gmail.com (Y.-M.N.); mhpark@chonnam.ac.kr (M.-H.P.); 2Department of Nuclear Medicine, Chonnam National University Medical School, 322 Seoyang-ro Hwasun-eup, Hwasun-gun, Gwangju 58128, Korea

**Keywords:** papillary thyroid carcinoma, lymph node dissection, skip metastases, thyroglobulin, recurrence

## Abstract

Skip lymph node (LN) metastases in papillary thyroid carcinoma (PTC) belong to N1b classification in the absence of central neck LN involvement. This study aimed to evaluate the predictive factors of skip metastases and their impact on recurrence in PTC patients with pN1b. A total of 334 PTC patients who underwent total thyroidectomy with LN dissection (central and lateral neck compartment) followed by radioactive iodine ablation were included. Patients with skip metastases tended to have a small primary tumor (≤1 cm) and single lateral neck level involvement. Tumor size ≤ 1 cm was an important predictive factor for skip metastases. Univariate analysis for recurrence showed that patients with a central LN ratio > 0.68, lateral LN ratio > 0.21, and stimulated thyroglobulin (Tg) levels > 7.3 ng/mL had shorter RFS (recurrence-free survival). The stimulated Tg level was associated with shorter RFS on multivariate analysis (>7.3 vs. ≤7.3 ng/mL; hazard ratio, 4.226; 95% confidence interval, 2.226−8.022; *p* < 0.001). Although patients with skip metastases tended to have a small primary tumor and lower burden of lateral neck LN involvement, there was no association between skip metastases and RFS in PTC with pN1b. Stimulated Tg level was a strong predictor of recurrence.

## 1. Introduction

Although papillary thyroid carcinoma (PTC) has the highest incidence among thyroid malignancies, its disease progression is known to be slower, irrespective of whether lymph node (LN) involvement is high (up to 80% at first diagnosis) [1]. It is generally acceptable that the lymphatic pathway of tumor dissemination in PTC is from the central to the ipsilateral or contralateral lateral compartments [2]. The preoperative detection of metastatic central LNs is more difficult than that of metastatic lateral LNs because central LNs are relatively small and adjoin the air-filled trachea or the thyroid gland itself. However, performing CND is necessary in patients with lateral neck LN metastases. Postoperative pathology has revealed that some patients who underwent lateral neck dissection had no central LN involvement. This spread pattern of LN metastases is referred to as skip metastases.

The reported prevalence of skip metastases ranges from 6.9% to 21.8% [3,4,5,6,7]. For the determination of skip metastases, compartment-oriented neck dissection should be an essential strategy in PTC with lateral neck LN metastases [8]. Recently, the cancer staging system indicated that the location of the involved LNs is not important [9]. In addition, the American Thyroid Association (ATA) guidelines focus on the number and size of the involved LNs, the extranodal extension, and clinical suspicion of LNs, rather than on the location of involved LN [10]. Nevertheless, recurrence-free survival (RFS) is lower in PTC patients with pN1b (LN involvement in the lateral compartment) than in those with pN1a (LN involvement in the central compartment) or N0 (no LN involvement) [11,12,13]. Patients with lateral neck LN metastases are classified as N1b classification regardless of the presence of central LN involvement. Only a few studies have investigated the impact of skip metastases on survival outcomes in PTC with pN1b. Hence, the aim of this study was to evaluate the predictive factors of skip metastases and to determine prognostic factors for recurrence in PTC patients who had undergone total thyroidectomy and CND associated with ipsilateral or bilateral lateral neck dissection, followed by radioactive iodine ablation (RIA).

## 2. Materials and Methods

### 2.1. Study Population

We reviewed the electronic medical records of patients who underwent thyroid surgery at our institution between January 2006 and December 2014. We collected data of patients with N1b classification who had undergone total thyroidectomy due to PTC. Patients aged less than 15 or over 80 years; those who underwent selective lateral neck dissection or berry picking; did not undergo CND; had second malignancies during the follow-up period; had persistent disease within six months after the initial surgery; had distant metastases at first diagnosis; underwent incomplete resection of the tumor; had insufficient information regarding stimulated thyroglobulin (Tg) levels or anti Tg- antibody levels, or tested positive for the anti-Tg antibody (≥60 IU/mL) were excluded (Figure 1). A total 334 patients with PTC who underwent total thyroidectomy and CND with ipsilateral or bilateral lateral neck dissection followed by RIA were enrolled in this study. This retrospective study was approved by the institutional review board of our institution, and the need for consent was waived.

### 2.2. Surgery

All primary tumors with lateral neck LNs were evaluated by ultrasonography (US) and fine needle aspiration cytology (FNAC) was conducted irrespective of tumor size. In cases of discordance between the result of FNAC and interpretation of US in suspicious lateral neck LN, intraoperative frozen biopsy sections were taken in 37 patients for making a decision regarding lateral neck dissection. Preoperative measurement of the Tg level in needle washout was conducted with FNAC in 54 patients. All PTC patients who were planned for thyroidectomy and lateral neck dissection underwent encompassing bilateral CND, irrespective of central LNs involvement. Bilateral CND refers to removal of the pre-laryngeal, pre- and paratracheal, and/or para-esophaeal LN. Resection of the upper mediastinal LN (level VII) was applied for patients with clinically evident LNs, either preoperatively or intraoperatively. Lateral neck dissection involved the removal of all LNs in the neck level of II-V with preservation of the spinal accessory nerve, sternocleidomastoid muscle, and internal jugular vein. The excision of LN in level I was performed in cases of suspicious LNs because these are rare events in PTC. Intraoperative neuro-monitoring was selectively used for patients who had a tumor that was located on the posterior surface of the thyroid gland and had suspicious invasion of the recurrent laryngeal nerve since 2013.

### 2.3. Postoperative Management

All patients were administered 50–100 mCi of I-131 for RIA at 2 or 3 months after surgery. More than 30 uIU/mL of thyroid stimulating hormone (TSH) after T4 withdrawal or recombinant human TSH injection was an essential requirement for RIA. An immunoradiometric assay (RIA Tg-plus, BRAHMS GmbH, Hennigsdorf, Germany) was used in determining stimulated Tg levels (lower limit of 0.2 ng/mL), and a radioimmunoassay (RIA anti-Tgn, BRAHMS GmbH, Hennigsdorf, Germany) was used in measuring anti-Tg antibody levels (lower limit of 20 U/mL). Stimulated Tg was measured before RIA. TSH suppression therapy was administered to enrolled patients during the follow-up period. Physical examination, thyroid function test, Tg and anti-Tg antibody level measurements, and neck US were conducted every 3–6 months for 5 years, and annually thereafter. In patients with increased Tg or anti-Tg antibody levels, irrespective of clear US findings, neck computed tomography (CT), whole body scan, or 18F-fluorodeoxyglucose positron emission tomography CT were additionally performed. Recurrence was only considered when structural disease was present in any lesion. All patients in this study underwent neck US within 6 months after the initial operation, to distinguish for persistent disease [14]. Loco-regional recurrence was defined as the presence of structural disease on imaging studies in the central and lateral neck compartments. FNAC was used to access suspicious loco-regional recurrence. Patients with loco-regional recurrence underwent reoperation and further RIA, whereas radioactive iodine therapy was considered as the first treatment option for patients with metastases to distant organs.

### 2.4. Statistics

Normally distributed continuous variables are presented as mean (± standard deviation) and compared using the independent sample *t*-test. Non-normally distributed continuous variables are expressed as median (interquartile range) and compared using Mann-Whitney U test. Categorical variables are shown as a value (present) and compared using Chi-square or Fisher’s exact tests. Multivariate logistic regression analysis was used for the prediction of skip metastases. RFS was defined as the time between the initial surgery and first identification of structural recurrence. To evaluate the risk of recurrence, the Cox proportional hazard model was used for univariate and multivariate regression analyses, and the results are presented as the hazard ratio (HR) and 95% confidence interval (CI). Multivariate analyses of the significant variables on univariate analyses were performed using backward elimination. The optimal cutoff value for the LN ratio (involved LNs divided by harvested LNs) and stimulated Tg levels were estimated using the receiver operating characteristic (ROC) curve. We used the log-rank test and Kaplan–Meier curves to calculate differences in RFS. All statistical analyses were performed using SPSS (version 23.0; IBM Inc., Armonk, NY, USA), and *p* < 0.05 was considered statistically significant.

## 3. Results

### 3.1. Patients’ Characteristics

Table 1 shows patients’ demographics. Among the 334 patients included in this study, 255 (76.3%) were less than 55 years old, and 95 (28.4%) were male. Microcarcinoma, which is a primary tumor ≤1 cm in size, was observed in 117 (35.0%) patients, whereas 95 (28.4%) patients showed gross extrathyroidal extension (ETE). Ipsilateral and bilateral lateral neck dissections were performed in 294 (88.0%) and 40 (12.0%) patients, respectively. Of all patients, 34 (10.2%) had skip metastases. During a median follow-up of 81 months (range, 15–154 months), recurrence occurred in 46 (13.8%) patients.

In the patients with recurrence, 13 had recurrence in the central compartment and five had recurrence in the central and ipsilateral lateral neck compartments. Additionally, 19 and eight patients had recurrence in ipsilateral and contralateral lateral neck compartments, respectively. Two patients developed distant metastases, one patient had only bone metastases, and the remaining patient had lung metastases with loco-regional recurrence.

### 3.2. Comparison of Patients with and without Skip Metastases

The mean age of patients with and without skip metastases were 48 and 45 years, respectively. Median tumor size was smaller in patients with skip metastases. The involvement of single-level metastases in the lateral compartment was more frequent in patients with skip metastases. There was no difference in age, sex, gross ETE, multifocality of tumors, bilaterality of tumors, and extent of lateral neck dissection according to the presence or absence of skip metastases (Table 2). Tumor size less than 1 cm was an important predictive factor for skip metastases in the logistic regression analyses (vs. >1 cm; odds ratio, 2.611; 95% confidence interval (CI), 1.273−5.357; *p* = 0.009) (Table 3).

### 3.3. Uni- and Multivariate Analysis According to Recurrence

Univariate analysis revealed that patients with central LN ratio > 0.68 (vs. ≤0.68; hazard ratio (HR), 2.831; 95% CI, 1.573−5.093; *p* = 0.001), lateral LN ratio > 0.21 (vs. ≤0.21; HR, 2.997; 95% CI, 1.522−5.901; *p* = 0.001), and stimulated Tg levels > 7.3 ng/mL (vs. ≤7.3 ng/mL; HR, 5.697; 95% CI, 3.102−10.462; *p* < 0.001) had worse RFS. In contrast, skip metastases were not associated with RFS. Moreover, age, sex, body mass index, tumor size, gross ETE, multifocality and bilaterality of tumors, extent of lateral neck dissection, lymphovascular invasion, and chronic lymphocytic thyroiditis had no impact on recurrence (Table 4 and Figure 2).

Only stimulated Tg levels >7.3 ng/mL were associated with poorer RFS, as shown by the multivariate analysis (vs. ≤7.3 ng/mL; HR, 4.226; 95% CI, 2.226−8.022; *p* < 0.001) (Table 5 and Figure 3).

The optimal cutoff value for the central LN ratio, lateral LN ratio, and stimulated Tg level was 0.68 (area under the curve (AUC), 0.626; sensibility, 58.7%; specificity, 68.1%), 0.21 (AUC, 0.633; sensibility, 76.1%; specificity, 51.0%), and 7.3 ng/mL (AUC, 0.763; sensibility, 65.2%; specificity, 77.8%) on ROC curve analysis, respectively (Figure 4).

## 4. Discussion

The present study was designed to evaluate the predictive factors of skip metastases and to analyze the prognostic factors for recurrence in patients who had undergone total thyroidectomy and bilateral CND with therapeutic ipsilateral or bilateral lateral neck dissection, followed by RIA during a median follow-up period of 81 months. Although only a small number of PTC patients with pN1b showed skip metastases (10.2%), this value is acceptable compared to the incidence of skip metastases in previous studies (ranging from 6.9% to 21.8%) [3,4,5,6,7]. Skip metastases were associated with lower aggressiveness such as small primary tumors and single lateral neck level involvement. However, there was no significant association in terms of longer RFS. Recurrence was predicted based on the central LN ratio, lateral LN ratio, and stimulated Tg levels in PTC patients with pN1b. Only stimulated Tg levels before RIA were found to be a strong prognostic factor for recurrence on multivariate analysis.

Cervical LN metastasis is commonly observed in PTC and is found during the initial operation in up to 60% of cases [15]. In a study with patients who had unilateral PTC and ipsilateral lateral neck LN involvement, prophylactic contralateral neck dissection (level III and IV) revealed occult LN metastasis in 36.5% of enrolled patients, irrespective of negative preoperative findings in the contralateral lateral compartment [16]. Nevertheless, performing prophylactic lateral neck dissection without evidence of result from FNAC or Tg levels in the needle washout was not recommended according to the recent guidelines [10,17]. Considering the general dissemination of LN metastases (from central to lateral compartment) in PTC, the decision for skip metastases depends on the status of central LN. Unfortunately, the identification of LN-related factors for predicting skip metastases is difficult in PTC with lateral neck LN metastases. Moreover, the detection rate of LN involvement in the central compartment is lower than that in the lateral compartment during preoperative evaluation [18]. Lu et al. suggested a scoring system for predicting the metastases of specific LN in the central compartment [19]. However, CND should be mandatory in PTC patients with lateral neck LN metastases to avoid persistent disease and false negative skip metastases. 

Even though active surveillance has become an alternative strategic option for patients with micro PTC, some patients with micro PTC had distant metastases at initial diagnosis [20]. Although recent ATA guidelines recommend that nodules ≥1 cm size with highly suspicious US findings need diagnostic FNAC to exclude or confirm malignancy, FNAC may be needed even if nodules ≤1 cm have evidence of ETE or suspicious metastatic lateral neck LNs. Some authors demonstrated that the tumor >5 mm in the upper pole of thyroid gland had a high likelihood of lateral LN metastases irrespective of micro PTC [21]. Similar to other previous studies [3,7], our study revealed that micro PTC was strong predictive factor for skip metastases. Moreover, primary tumor located in the upper thyroid gland was associated with skip metastases because of the assumption of lymphatic drainage running close to the superior thyroid artery [22]. Consequentially, some authors demonstrated that the involvement of levels II was commonly observed in PTC patients with skip metastases [23]. This present study showed that the involvement of lateral neck level in patients with skip metastases was present in 26 (76.5%) in level III, 17 (50.0%) in level IV, 11 (32.4%) in level II, and two patients (5.9%) in level V. Compared to previous studies, it was hard to reveal the relationship between the tumor location and skip metastases because some cases could not be defined by the tumor location in these enrolled patients. That is why the involvement of level III and IV were commonly observed in patients with skip metastases. We are planning further research regarding the association between tumor location and skip metastases in patients with distinguishable tumor locations. 

Among the LN-related factors, the LN ratio is widely used for supplementing the numerical value of harvested or metastatic LNs in TNM staging. Various cut-off levels associated with recurrence in PTC have been reported [24,25]. One study reported that the LN ratio in the central compartment, rather than in the lateral compartment, is an independent risk factor for loco-regional recurrence in patients with PTC who have undergone lateral neck dissection [26]. In contrast, the LN ratio in the lateral compartment is useful for predicting loco-regional and any lesion recurrence [27,28]. The present study calculated the optimal cut-off point values of central and lateral LN ratio for recurrence using ROC curve analysis. However, we found that the individual LN ratio was associated with improved RFS only by univariate analysis. There has been no consensus on the appropriate number of resected LNs in thyroid cancer. In order to utilize the LN ratio for the supplementation of numbers of metastatic LN, encompassing compartment-oriented neck dissection including CND should be performed in patients with lateral neck LN metastases. The lack of thorough inspection of LNs in the central compartment may provide false negative results for skip metastases and may impact RIA and the intensity of surveillance. 

Clinical response after surgery and RIA in differentiated thyroid cancer is classified according to imaging findings, and the levels of Tg and anti-Tg antibody. Patients with PTC who undergo lateral neck dissection are classified as having intermediate or high risk for structural recurrence. Therefore, most PTC patients with pN1b need postoperative RIA. Stimulated Tg levels are useful as a biochemical marker to predict disease-free status in patients with PTC [29]. In a study with 452 patients with differentiated thyroid cancer who had undergone total thyroidectomy with or without cervical LN dissection, 82 patients with pre-ablation Tg levels <1 ng/mL had no structural recurrence, whereas 42% of 197 patients with pre-ablation Tg levels >10 ng/mL developed recurrence [30]. Patients with abnormal Tg value (non-stimulated Tg ≥ 1 ng/mL or stimulated Tg ≥ 10 ng/mL) and without localizable disease were classified as having a biochemical incomplete response according to the risk stratification of ATA guidelines [10]. However, elevated levels of Tg are not always considered as having structural recurrence. One study demonstrated that more than 30% of patients with biochemical incomplete response after total thyroidectomy and RAI ablation were shifted to spontaneous conversion without evidence of structural disease [31]. Another study reported that close to half of patients with stimulated Tg ≥ 10 ng/mL during the first 12–24 months after total thyroidectomy and radioactive iodine ablation had no structural disease [32]. The present study found that 78 patients (27%) without structural recurrence had non-stimulated Tg ≥ 1 ng/mL and 21 patients (45.7%) with structural recurrence had non-stimulated Tg < 1 ng/mL. Therefore, our study gave weight to structural recurrence regardless of the level of non-stimulated Tg. If complete resection of the thyroid gland and the compartment-oriented LNs is achieved, the level of stimulated Tg before RIA can be an excellent predictive factor for recurrence. Thus, the meticulous preoperative evaluation of the lateral neck compartment is required in patients with PTC to avoid missing small suspicious LN in the lateral neck. 

This study has several limitations. Firstly, the study design was retrospective in nature, despite maintaining the same diagnostic and surgical concepts at a single institution. Secondly, we did not consider a biochemically incomplete or indeterminate response as a recurrence. Finally, information concerning LN-related factors, such as the maximal size and extranodal extension of metastatic LN, were not acquired in patients who were included at the beginning of this study. Thus, further prospective protocols with accurate and reliable individual information are needed to support our results.

## 5. Conclusions

The assessment of central LN is necessary to evaluate skip metastases in PTC with lateral neck LN metastases. We found that patients with skip metastases tend to have microcarcinoma and a lower burden of lateral neck LN involvement. Although there was no association between skip metastases and RFS, CND should be performed routinely in PTC patients with lateral neck LN metastases. Structural recurrence was predicted by stimulated Tg levels before RIA in PTC patients with pN1b. Thus, close monitoring using imaging modalities and flexible interval follow-up are necessary according to the level of stimulated Tg in PTC patients with pN1b.

## Figures and Tables

**Figure 1 biomedicines-10-00179-f001:**
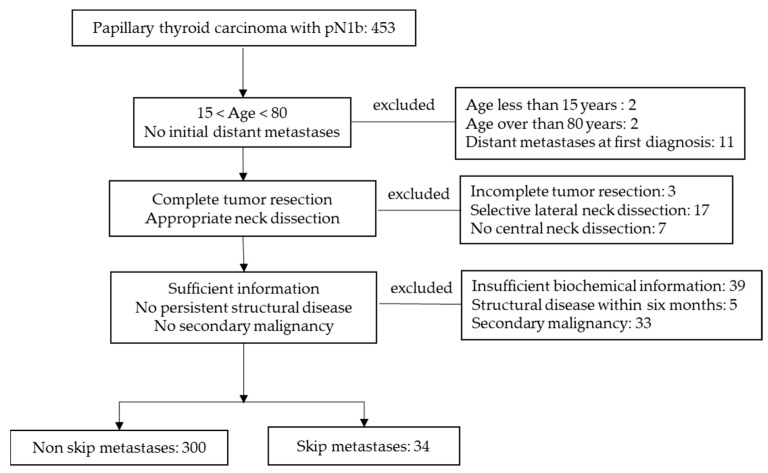
Flowchart of study population.

**Figure 2 biomedicines-10-00179-f002:**
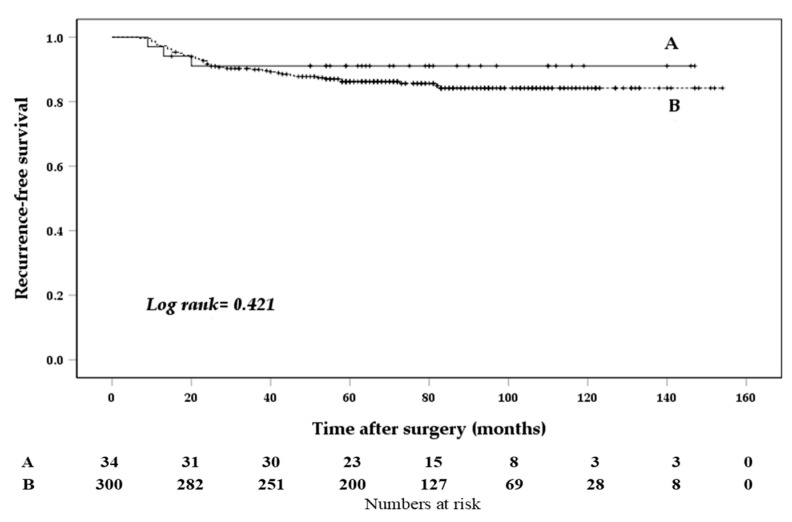
Kaplan-Meier curve according to the presence and absence of skip metastases. Skip metastases (**A**), Non-skip metastases (**B**).

**Figure 3 biomedicines-10-00179-f003:**
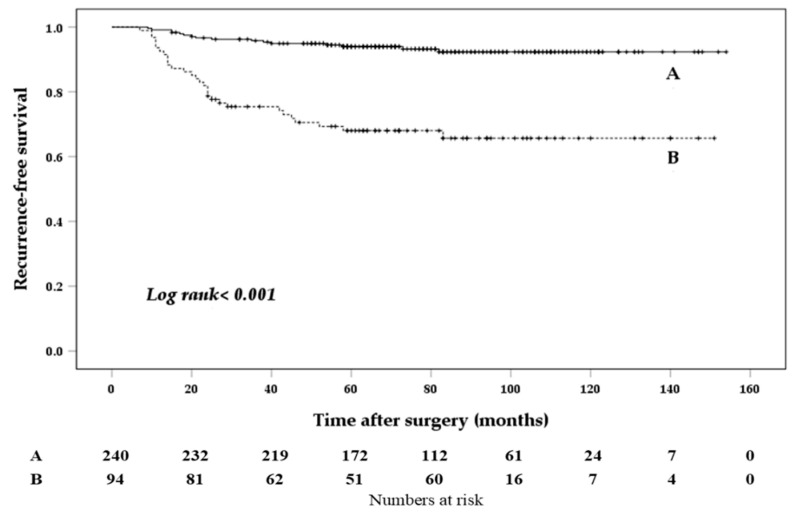
Kaplan-Meier curve according to the level of stimulated thyroglobulin (Tg). Tg ≤ 7.3 ng/mL (**A**), Tg > 7.3 ng/mL (**B**).

**Figure 4 biomedicines-10-00179-f004:**
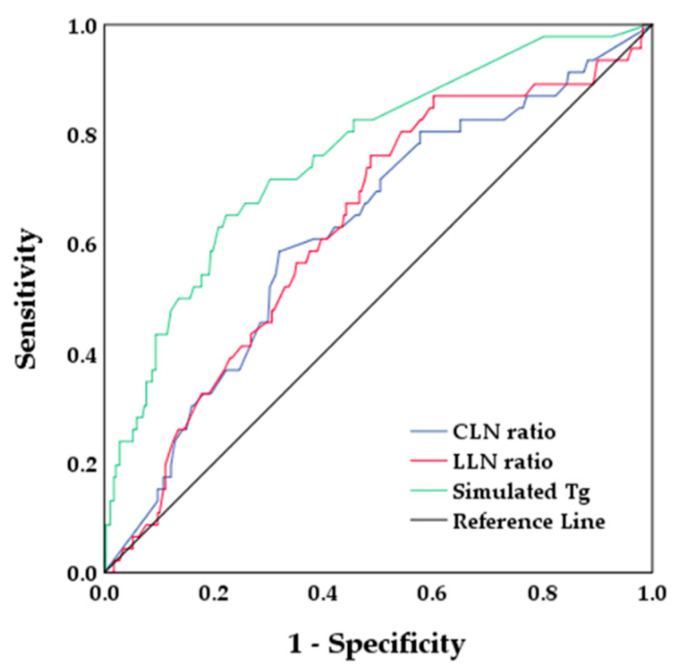
Receiver operating characteristic curves. Area under the curve of CLN ratio, LLN ratio, and stimulated Tg was 0.626, 0.633, and 0.763, respectively. CLN, central lymph node; LLN, lateral lymph node; Tg, thyroglobulin.

**Table 1 biomedicines-10-00179-t001:** Patients’ demographics.

Variables	Value, Number (Percent)
Total	334 (100)
Age	
<55 years/≥55 years	255 (76.3)/79 (23.7)
Mean (standard deviation)	45.3 years (13.1)
Sex	
Female/Male	239 (71.6)/95 (28.4)
Tumor size	
≤1 cm/>1 cm	117 (35.0)/217 (65.0)
Median (interquartile range)	1.4 cm (1.1)
T classification *	
T1a/T1b	105 (31.4)/96 (28.7)
T2	35 (10.5)
T3a/T3b	3 (0.9)/44 (13.2)
T4a	51 (15.3)
TNM stage *	
I/II/III	255 (76.3)/65 (19.5)/14 (4.2)
pN1b and rM0	332 (99.4)
Gross ETE *	95 (28.4)
Multifocality	113 (33.8)
Unilateral/Bilateral	17 (5.1)/96 (28.7)
Extent of lateral neck dissection	
Ipsilateral/Bilateral	294 (88.0)/40 (12.0)
Skip metastases	34 (10.2)
Median follow-up (range)	81 months (15−154 months)

ETE, extrathyroidal extension. * Based on the 8th edition of the AJCC/TNM staging system.

**Table 2 biomedicines-10-00179-t002:** Comparison of patients with and without skip metastases.

Variables	Skip Metastases		*p* Value
	Absence	Presence	
Age			0.404
<55 years	231 (77.0)	24 (70.6)	
≥55 years	69 (23.0)	10 (29.4)	
Mean ± standard deviation (years)	45.0 ± 13.2	48.0 ± 11.2	0.216
Sex			0.141
Female	211 (70.3)	28 (82.4)	
Male	89 (29.7)	6 (17.6)	
Tumor size			0.013
≤1 cm	98 (32.7)	19 (55.9)	
>1 cm	202 (67.3)	15 (44.1)	
Median (interquartile range)	1.4 (1.0)	0.8 (1.2)	<0.001
Gross ETE	89 (29.7)	6 (17.6)	0.164
Multifocality	105 (35.0)	8 (23.5)	0.180
Bilaterality	89 (29.7)	7 (20.6)	0.268
Extent of lateral neck dissection			0.401
Ipsilateral	262 (87.3)	32 (94.1)	
Bilateral	38 (12.7)	2 (5.9)	
TNM stage			0.404
I	231 (77.0)	24 (70.6)	
II and III	69 (23.0)	10 (29.4)	
Number of harvested LN			
Central LN, median (interquartile range)	8.0 (6)	5.0 (5)	<0.001
Lateral LN, median (interquartile range)	19.0 (12)	19 (15)	0.953
Number of metastatic LN			
Central LN, median (interquartile range)	4.0 (4)	0	<0.001
Lateral LN, median (interquartile range)	5.0 (4)	2.5 (2)	<0.001
Single lateral level involvement	64 (21.3)	19 (55.9)	<0.001
Recurrence	43 (14.3)	3 (8.8)	0.598
Total	300 (100)	34 (100)	

ETE, extrathyroidal extension; LN, lymph node.

**Table 3 biomedicines-10-00179-t003:** Multivariate logistic regression analyses for the prediction of skip metastases.

Variables (Reference)		OR (95% CI)	*p* Value
Age (<55 years)	≥55 years	1.360 (0.608−3.039)	0.454
Sex (Female)	male	0.504 (0.200−1.268)	0.145
Tumor size (>1 cm)	≤1 cm	2.611 (1.273−5.357)	0.009
Gross ETE (absence)	presence	0.674 (0.257−1.768)	0.423
Multifocality (absence)	presence	0.594 (0.257−1.373)	0.224
Bilaterality (absence)	presence	1.372 (0.154−12.212)	0.777
Extent of lateral neck dissection (ipsilateral)	bilateral	0.765 (0.163−3.585)	0.734

CI, confidence interval; ETE, extrathyroidal extension; OR, odds ratio.

**Table 4 biomedicines-10-00179-t004:** Univariate analysis related to recurrence.

Variables (Reference)		HR (95% CI)	*p* Value
Age (<55 years)	≥55 years	1.250 (0.658−2.374)	0.496
Sex (female)	male	0.887 (0.459−1.713)	0.722
Tumor size (≤1 cm)	>1 cm	1.423 (0.749−2.704)	0.281
Gross ETE (absence)	presence	1.572 (0.864−2.862)	0.139
Multifocality (absence)	presence	1.250 (0.691−2.261)	0.460
Bilaterality (absence)	presence	1.181 (0.637−2.188)	0.598
Extent of lateral neck dissection (ipsilateral)	bilateral	1.408 (0.630−3.148)	0.405
Skip metastases (absence)	presence	0.622 (0.193−2.004)	0.426
Central LN ratio (≤0.68)	>0.68	2.831 (1.573−5.093)	0.001
Lateral LN ratio (≤0.21)	>0.21	2.997 (1.522−5.901)	0.001
TNM stage (I)	(II and III)	1.250 (0.658−2.374)	0.496
Stimulated thyroglobulin (≤7.3 ng/mL)	>7.3 ng/mL	5.697 (3.102−10.462)	<0.001

CI, confidence interval; ETE, extrathyroidal extension; HR, hazard ratio; LN, lymph node.

**Table 5 biomedicines-10-00179-t005:** Multivariate analysis related to recurrence.

Variables (Reference)		HR (95% CI)	*p* Value
Central LN ratio (≤0.68)	>0.68	1.745 (0.947−3.216)	0.074
Lateral LN ratio (≤0.21)	>0.21	1.832 (0.906−3.704)	0.092
Stimulated thyroglobulin (≤7.3 ng/mL)	>7.3 ng/mL	4.226 (2.226−8.022)	<0.001

CI, confidence interval; HR, hazard ratio; LN, lymph node.

## Data Availability

The datasets generated during and/or analyzed during the current study are available from the corresponding author on reasonable request.

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
