# Peer review of "Predictive Factors for Skip Lymph Node Metastasis and Their Implication on Recurrence in Papillary Thyroid Carcinoma"

_biomedicines, 2022, doi:10.3390/biomedicines10010179_

Round 1

Reviewer 1 Report

Abstract: redundant sentence at the end: "stimulated Tg level was a strong predictor of recurrence." It is already included in the abstract two lines above.

Methods: "stimulated Tg": could you give more explanation on the stimulating agent and the procedure?

Recurrence of PTC : how many among the initial 334 cases? By which method (an increase of stimulated Tg was sufficient to declare a recurrence? ...)? Within which time-to-event delay?

Reviewer 2 Report

The manuscript „Predictive factors for skip lymph node metastasis and their implication on recurrence in papillary thyroid carcinoma” is very interesting  observational study.  The methodology is sufficient and good described. The results were interesting presented. The introduction and discussion need improvement, because in 2021  were more than 300 publications  about lymph node metastases and recurrence in thyroid papillary carcinoma released. The pivotal clinical relevance have for example:

Lu KN, Zhang Y, Da JY, Zhou TH, Zhao LQ, Peng Y, Pan G, Shi JJ, Zhou L, Ni YQ, Luo DC. A Novel Scoring System for Predicting the Metastases of Posterior Right Recurrent Laryngeal Nerve Lymph Node Involvement in Patients With Papillary Thyroid Carcinoma by Preoperative Ultrasound. Front Endocrinol (Lausanne). 2021 Aug 31;12:738138.

Zhang X, Chen W, Fang Q, Fan J, Feng L, Guo L, Liu S, Ge H, Du W. Lateral Lymph Node Metastases in T1a Papillary Thyroid Carcinoma: Stratification by Tumor Location and Size. Front Endocrinol (Lausanne). 2021 Jul 15;12:716082.

Sudoko CK, Jenks CM, Bauer AJ, Isaza A, Mostoufi-Moab S, Surrey LF, Bhatti TR, Franco A, Adzick NS, Kazahaya K. Thyroid Lobectomy for T1 Papillary Thyroid Carcinoma in Pediatric Patients. JAMA Otolaryngol Head Neck Surg. 2021 Nov 1;147(11):943-950.

Lončar I, van Dijk SPJ, Metman MJH, Lin JF, Kruijff S, Peeters RP, Engelsman AF, van Ginhoven TM. Active Surveillance for Papillary Thyroid Microcarcinoma in a Population with Restrictive Diagnostic Workup Strategies. Thyroid. 2021 Aug;31(8):1219-1225.

In this situation introduction and discussion need revision  and in references should by included latest publications.

Author Response

Comments and Suggestions for Authors

Thank you for your insightful comments. We are pleased that you enjoyed reading our paper.

The manuscript „Predictive factors for skip lymph node metastasis and their implication on recurrence in papillary thyroid carcinoma” is very interesting  observational study.  The methodology is sufficient and good described. The results were interesting presented. The introduction and discussion need improvement, because in 2021  were more than 300 publications  about lymph node metastases and recurrence in thyroid papillary carcinoma released. The pivotal clinical relevance have for example:

Lu KN, Zhang Y, Da JY, Zhou TH, Zhao LQ, Peng Y, Pan G, Shi JJ, Zhou L, Ni YQ, Luo DC. A Novel Scoring System for Predicting the Metastases of Posterior Right Recurrent Laryngeal Nerve Lymph Node Involvement in Patients With Papillary Thyroid Carcinoma by Preoperative Ultrasound. Front Endocrinol (Lausanne). 2021 Aug 31;12:738138.

Zhang X, Chen W, Fang Q, Fan J, Feng L, Guo L, Liu S, Ge H, Du W. Lateral Lymph Node Metastases in T1a Papillary Thyroid Carcinoma: Stratification by Tumor Location and Size. Front Endocrinol (Lausanne). 2021 Jul 15;12:716082.

Sudoko CK, Jenks CM, Bauer AJ, Isaza A, Mostoufi-Moab S, Surrey LF, Bhatti TR, Franco A, Adzick NS, Kazahaya K. Thyroid Lobectomy for T1 Papillary Thyroid Carcinoma in Pediatric Patients. JAMA Otolaryngol Head Neck Surg. 2021 Nov 1;147(11):943-950.

Lončar I, van Dijk SPJ, Metman MJH, Lin JF, Kruijff S, Peeters RP, Engelsman AF, van Ginhoven TM. Active Surveillance for Papillary Thyroid Microcarcinoma in a Population with Restrictive Diagnostic Workup Strategies. Thyroid. 2021 Aug;31(8):1219-1225.

In this situation introduction and discussion need revision and in references should by included latest publications.

Reply: Thank you for your comments. We had modified introduction and discussion section and included latest publication.

Again thank you for your time reading the paper.Reviewer reports:

Leyre Lorente Poch (Reviewer 1): The study assesses the predictive factors for recurrence after total thyroidectomy plus bilateral prophylactic central neck dissection for papillary thyroid cancer, a still ongoing clinical topic of controversy.

I would like to make some comments:
1- The title is not fully in agreement with the content of this study. It may benefit of rewriting it since the authors are not analyzing the effectiveness of prophylactic central compartment neck dissection. Risk factors for recurrence are assessed: among them, LN ratio more than 5 and more than 3 positive LN are shown to be predictive factor for recurrence.
2- Background:
There is a missing comprehensive meta-analysis published by the ESES wherein prophylactic neck dissection is extensively analyzed (Sancho, J. J., Lennard, T. W. J., Paunovic, I., Triponez, F., & Sitges-Serra, A. (2014). Prophylactic central neck disection in papillary thyroid cancer: a consensus report of the European Society of Endocrine Surgeons (ESES). Langenbeck's archives of surgery, 399(2), 155-163). The authors may consider including it as a reference.
It is true that there is a scarcity of level 1 evidence or direct evidence of benefit in relation to prophylactic central compartment dissection but there have been great amount of studies dealing with the issue.

3- Methods:
a) Patients without CND were excluded from the analysis: may the authors consider using them as a control group to compare if there are different rates of recurrence?
b) The authors must specified whether all surgeries were performed by the same team of surgeons.
c) Bethesda V and VI were included: there is no mention of which percentage of patients finally had papillary carcinoma proved by the biopsy. The authors should include this information.
d) In line 38 distinction between transient and permanent applies to which complication? It must be clarified.
e) When was PTH and calcium measured? Do the authors have any protocol for postoperative hypocalcaemia in your center?
f) It is of utmost importance to clarify whether the end-point is structural local recurrence or structural distant recurrence.

4- Results: According to these results, patients having more than 50% of positive LN in the CND sample had 5 times more recurrence (of note that more than 40% of patients had 4 or less harvested LN). LF ratio was the predicting factor which had more impact on the incidence of recurrence. I would suggest the authors developing a little more the interpretation of the results, since data are already reflected in the tables.
a) Table 2 and 3: Adding in the title "Univariate Cox Regression of recurrence" "Multivariate Cox Regression of recurrence" would enhance the clarity of the table. Additionally Authors should write "HR (Hazard ratios)" instead of "Exp(B)".
b) Table 4: according to these data, among 1082 patients who underwent total thyroidectomy there is no wound infection. Such an outstanding result mandates to consider re-definition of criteria for wound infection.
c) Line 7 (page 8): the age range is reflected to be from 7 to 75 years. One of the exclusion criteria was patients younger than 15 years old. Please clarify.

5- Discussion and Conclusion:
The Authors should explain which is the reasoning to advise bilateral CND instead of ipsilateral CND? Do the Authors have data proving whether positive nodes were on the contralateral or same side of the primary tumor? This data would be interesting.
From the 62 structural recurrences of this study, more than 30% occurred in the central neck compartment despite having performed prophylactic bilateral neck dissection. Maybe additional conclusion would be to be focus in a thorough dissection of compartment VI. Data regarding in which side (right or left) and which group (pre-traqueal, para-esophageal, Delphian node…) was the recurrence happening would be interesting to know.

Carmela De Crea, MD PhD (Reviewer 2): BSUR-D-18-00321

Efficacy of prophylactic central lymph node dissection for papillary thyroid carcinoma with pathological N1a Young Jae Ryu, M.D.; Jin Seong Cho, M.D.; Min Ho Park, M.D.; Jung Han Yoon, M.D.

Dr. Ryu and colleagues have submitted a retrospective patient's series of 1082 patients who underwent total thyroidectomy and prophylactic central neck node dissection between January 2004 and December 2012. The aim of the study was to evaluate the predictive factors of recurrence in cN0 papillary thyroid carcinoma (PTC) patients with pathologic N1a. The cN0 patients without lymph node involvement at the pathologic examination were excluded. At a median follow-up of 78 months, the rate of recurrence was 5.7%.
The role and the extension of prophylactic central neck node dissection in PTC patients is still controversial and represent an interesting topic in the current literature. In addition, the study evaluated a quite large patient series operated on in a single Institution and may provide some useful clinical information.
However, this is a retrospective study and this represent a limitation as the authors stated. Moreover, some more significant limitations are inherent in the study and should be addresses in order to strengthen the paper and insure that the conclusions offered are supported by the data.

1. Since in the study the cN0 patients without lymph node involvement at the pathologic examination were excluded, the authors should specify the overall rate of PTC N1a patients. In addition, it would be interesting to know the rate of micrometases in central neck node.
2. In the methods section, the authors assessed that comprehensive central LN dissection included level VI and VII. However, in 41.3% of patients the number of harvested central neck node was ≤ 4. The authors should better clarify this point specifying the mean number (and the range) of excised central neck node, in order to assess the adequacy of surgical resections. Indeed, in the manuscript, no data are available regarding the postoperative basal and/or stimulated thyroglobulin, and this is a major drawback, as the authors stated in the discussion section.
3. In the reported patients series there were respectively 119 T3b and 57 T4a. Did the authors checked the lateral neck node (e.g. intraoperative frozen section) in these cases? This data would be relevant considering that more than 50% of the recurrences of the present series were in the lateral neck node.
4. Between the 19 patients that had local recurrence in the "operative bed or central neck compartment", the authors should specify how many were the nodal recurrences.
5. Can the authors specify the mean time to recurrence?
6. In the methods section the authors define postoperative hypoparathyroidism following very strict criteria. The rate of postoperative transient hypoparathyroidism reported (5.8%), seems quite low, for a patients series underwent to total thyroidectomy and comprehensive central neck node dissection. Could the authors further specify the postoperative protocol followed, with particular refer to the timing of postoperative PTH measurement.
7. Did the authors submit the patients to routinely pre- and postoperative fibrolaryngoscopy?
8. More than 90% of the patients of the series underwent postoperative RAI. However, follow-up data regarding the completeness of surgical resection are lacking and this is a major limitation of the study. The authors should further discuss such high percentage of postoperative RAI.
9. In unifocal cN0 PTC, ipsilateral central compartment node dissection with frozen section examination has been demonstrated as a valid alternative to prophylactic bilateral central neck node dissection, since it allows accurate staging and may reduce morbidity. Indeed, considered the topic, the approach should be cited in the discussion section. Since 60.3% of the patients had papillary microcarcinoma and overall rate of bilateral multifocality reported was 22.4%, did the authors had any experience with this approach?
10. Even if the references list refers to adequate and powerful literature, it would be advisable to add a more update data widely published in the present endocrine surgery literature.Reviewer reports:

Leyre Lorente Poch (Reviewer 1): The study assesses the predictive factors for recurrence after total thyroidectomy plus bilateral prophylactic central neck dissection for papillary thyroid cancer, a still ongoing clinical topic of controversy.

I would like to make some comments:
1- The title is not fully in agreement with the content of this study. It may benefit of rewriting it since the authors are not analyzing the effectiveness of prophylactic central compartment neck dissection. Risk factors for recurrence are assessed: among them, LN ratio more than 5 and more than 3 positive LN are shown to be predictive factor for recurrence.
2- Background:
There is a missing comprehensive meta-analysis published by the ESES wherein prophylactic neck dissection is extensively analyzed (Sancho, J. J., Lennard, T. W. J., Paunovic, I., Triponez, F., & Sitges-Serra, A. (2014). Prophylactic central neck disection in papillary thyroid cancer: a consensus report of the European Society of Endocrine Surgeons (ESES). Langenbeck's archives of surgery, 399(2), 155-163). The authors may consider including it as a reference.
It is true that there is a scarcity of level 1 evidence or direct evidence of benefit in relation to prophylactic central compartment dissection but there have been great amount of studies dealing with the issue.

3- Methods:
a) Patients without CND were excluded from the analysis: may the authors consider using them as a control group to compare if there are different rates of recurrence?
b) The authors must specified whether all surgeries were performed by the same team of surgeons.
c) Bethesda V and VI were included: there is no mention of which percentage of patients finally had papillary carcinoma proved by the biopsy. The authors should include this information.
d) In line 38 distinction between transient and permanent applies to which complication? It must be clarified.
e) When was PTH and calcium measured? Do the authors have any protocol for postoperative hypocalcaemia in your center?
f) It is of utmost importance to clarify whether the end-point is structural local recurrence or structural distant recurrence.

4- Results: According to these results, patients having more than 50% of positive LN in the CND sample had 5 times more recurrence (of note that more than 40% of patients had 4 or less harvested LN). LF ratio was the predicting factor which had more impact on the incidence of recurrence. I would suggest the authors developing a little more the interpretation of the results, since data are already reflected in the tables.
a) Table 2 and 3: Adding in the title "Univariate Cox Regression of recurrence" "Multivariate Cox Regression of recurrence" would enhance the clarity of the table. Additionally Authors should write "HR (Hazard ratios)" instead of "Exp(B)".
b) Table 4: according to these data, among 1082 patients who underwent total thyroidectomy there is no wound infection. Such an outstanding result mandates to consider re-definition of criteria for wound infection.
c) Line 7 (page 8): the age range is reflected to be from 7 to 75 years. One of the exclusion criteria was patients younger than 15 years old. Please clarify.

5- Discussion and Conclusion:
The Authors should explain which is the reasoning to advise bilateral CND instead of ipsilateral CND? Do the Authors have data proving whether positive nodes were on the contralateral or same side of the primary tumor? This data would be interesting.
From the 62 structural recurrences of this study, more than 30% occurred in the central neck compartment despite having performed prophylactic bilateral neck dissection. Maybe additional conclusion would be to be focus in a thorough dissection of compartment VI. Data regarding in which side (right or left) and which group (pre-traqueal, para-esophageal, Delphian node…) was the recurrence happening would be interesting to know.

Carmela De Crea, MD PhD (Reviewer 2): BSUR-D-18-00321

Efficacy of prophylactic central lymph node dissection for papillary thyroid carcinoma with pathological N1a Young Jae Ryu, M.D.; Jin Seong Cho, M.D.; Min Ho Park, M.D.; Jung Han Yoon, M.D.

Dr. Ryu and colleagues have submitted a retrospective patient's series of 1082 patients who underwent total thyroidectomy and prophylactic central neck node dissection between January 2004 and December 2012. The aim of the study was to evaluate the predictive factors of recurrence in cN0 papillary thyroid carcinoma (PTC) patients with pathologic N1a. The cN0 patients without lymph node involvement at the pathologic examination were excluded. At a median follow-up of 78 months, the rate of recurrence was 5.7%.
The role and the extension of prophylactic central neck node dissection in PTC patients is still controversial and represent an interesting topic in the current literature. In addition, the study evaluated a quite large patient series operated on in a single Institution and may provide some useful clinical information.
However, this is a retrospective study and this represent a limitation as the authors stated. Moreover, some more significant limitations are inherent in the study and should be addresses in order to strengthen the paper and insure that the conclusions offered are supported by the data.

1. Since in the study the cN0 patients without lymph node involvement at the pathologic examination were excluded, the authors should specify the overall rate of PTC N1a patients. In addition, it would be interesting to know the rate of micrometases in central neck node.
2. In the methods section, the authors assessed that comprehensive central LN dissection included level VI and VII. However, in 41.3% of patients the number of harvested central neck node was ≤ 4. The authors should better clarify this point specifying the mean number (and the range) of excised central neck node, in order to assess the adequacy of surgical resections. Indeed, in the manuscript, no data are available regarding the postoperative basal and/or stimulated thyroglobulin, and this is a major drawback, as the authors stated in the discussion section.
3. In the reported patients series there were respectively 119 T3b and 57 T4a. Did the authors checked the lateral neck node (e.g. intraoperative frozen section) in these cases? This data would be relevant considering that more than 50% of the recurrences of the present series were in the lateral neck node.
4. Between the 19 patients that had local recurrence in the "operative bed or central neck compartment", the authors should specify how many were the nodal recurrences.
5. Can the authors specify the mean time to recurrence?
6. In the methods section the authors define postoperative hypoparathyroidism following very strict criteria. The rate of postoperative transient hypoparathyroidism reported (5.8%), seems quite low, for a patients series underwent to total thyroidectomy and comprehensive central neck node dissection. Could the authors further specify the postoperative protocol followed, with particular refer to the timing of postoperative PTH measurement.
7. Did the authors submit the patients to routinely pre- and postoperative fibrolaryngoscopy?
8. More than 90% of the patients of the series underwent postoperative RAI. However, follow-up data regarding the completeness of surgical resection are lacking and this is a major limitation of the study. The authors should further discuss such high percentage of postoperative RAI.
9. In unifocal cN0 PTC, ipsilateral central compartment node dissection with frozen section examination has been demonstrated as a valid alternative to prophylactic bilateral central neck node dissection, since it allows accurate staging and may reduce morbidity. Indeed, considered the topic, the approach should be cited in the discussion section. Since 60.3% of the patients had papillary microcarcinoma and overall rate of bilateral multifocality reported was 22.4%, did the authors had any experience with this approach?
10. Even if the references list refers to adequate and powerful literature, it would be advisable to add a more update data widely published in the present endocrine surgery literature.

Round 2

Reviewer 2 Report

The correction of manuscript is sufficient. I accept the current form the manuscript .